# Co-occurrence of diabetes and depression in the U.S.

**Maria L. Alva** *

Massive Data Institute, Georgetown University, Washington, DC, United States of America

* mla72@georgetown.edu

## Abstract

Evidence exists that depression interacts with physical illness to amplify the impact of chronic conditions like diabetes. The co-occurrence of these two conditions leads to worse health outcomes and higher healthcare costs. This study seeks to understand what demographic and socio-economic indicators can be used to predict co-occurrence at both the state and the individual level. Diabetes and depression are modeled as a bivariate normal distribution using data from the Behavioral Risk Factor Surveillance System 2016–2017 cohorts. The tetrachoric (latent) correlation between diabetes and depression is 17.2% and statistically significant, however the likelihood of any person being diagnosed with both conditions is small—as high as 4.3% (Arizona) and as low as 2.3% (Utah). We find that demographic characteristics (sex, age, and race) operate in opposite directions in predicting diabetes and depression diagnosis. Behavioral indicators (BMI≥30, smoking, and exercise); and life outcomes, (schooling attainment, marital and veteran status) work in the same direction to produce co-occurrence and as such are more powerful predictors of co-occurrence than demographic characteristics. It is important to have a rapid and efficient instrument to diagnoses co-occurrence. Simple questions about lifestyle choices, educational attainment and family life could help bridge the gap between primary care and psychological services with beneficial spillovers for patient-doctor communication.

## Introduction

During 2013–2016, 8.1% of Americans aged 20 and over reported having a depressive symptom in a given 2-week period [1]. Some people with MDD never get diagnosed, either because they do not seek care or because they are misdiagnosed. Depression is a common mental health disorder and there is growing evidence that is also significantly under-diagnosed. [2,3]. 9.4% of Americans have been diagnosed with diabetes in 2016 [4]. Just like depression, diabetes prevalence is also anticipated to grow, with estimates suggesting that the proportion of the population affected by diabetes will at least double by 2050 [5]. Currently, there is little funding on diabetes prevention yet, 1 in 4 health care dollars is spent to combat diabetes and its consequences [6]. While we lack accurate information about total and per capita cost of depression, there is evidence that comparatively, little funding has been available historically to combat depression and mental health in general. On average, states spend approximately 2% of their health dollars to the broad spectrum of mental health problems [7].

**Data Availability Statement:** Data are available to the public free of charge from the BRFSS web page.

**Funding:** The author received no specific funding for this work.

**Competing interests:** The author declares that no competing interests exist.

Meta-analyses have shown that depression is twice as prevalent among persons with diabetes as it is among persons without diabetes [8]. Depression has substantial economic repercussions when concurrent with other chronic illnesses. A study using data from the 2004–2011 Medical Expenditure Panel Survey (MEPS), a nationally representative estimate of healthcare expenditures, showed that the average medical cost for patients with diabetes and no depression was $10,016 (95% CI 9589–10,442), and with symptomatic depression was $20,105 (95% CI 18,103–22,106) [9]. The burden of this co-occurrence goes beyond the health care system. For example, depression is associated with a fourfold increase in 20+ days of reduced household work [10]. Evidence exists that depression interacts with physical illness to amplify the impact of other conditions and vice versa [11]. It increases adverse health behaviors like obesity, sedentary lifestyle, and substance abuse, which are risk factors for cardiovascular disease [12,13]. Depression is also associated with increased mortality post-myocardial infarction (Cox model hazard ratio for 6-month mortality associated with depression: 5.74 (95% CI: 4.61–6.87) p = .0006 [14]). Depression leads to decreased self-care and adherence to medical treatment, which adversely influences expectations of efficacy of treatment by reducing cognitive functioning and memory and reducing energy to exercise and follow regimens like checking blood glucose. Compared to nondepressed patients, patients with depression are significantly more likely to be non-adherent with medical treatment recommendations [15].

Because of all the above reasons, managing the co-occurrence of depressive disorders and chronic diseases, like diabetes, seems vital to health care delivery. Despite the growing recognition of the prevalence and importance of depressive disorders, there is little research examining factors that can predict the co-occurrence of depression with specific major chronic conditions and on how this knowledge could be used for prevention and treatment of populations at risk.

The goal of this study is to evaluate demographic and socio-economic indicators associated with both depression and diabetes at the macro (across-states) and micro level (individuals) by examining the likelihood of having:

i.  depression with diabetes

ii.  depression without diabetes

iii.  diabetes without depression

The policy-relevant questions this exercise seeks to answer are twofold: (1) what demographic and socioeconomic characteristics can help us identify state-level hotspots, i.e., are there relevant differences in the probability of being diagnosed with diabetes and depressions across states? And (2) can we help profile who might be more likely to experience co-occurrence of diabetes and depressions to target resources in a more targeted manner and in accordance to needs?

## Methods

### Data

This analysis uses publicly released data from the Behavioral Risk Factor Surveillance System (BRFSS), representing the 2016–2017 cohort. BRFSS is a representative health survey from non-institutionalized civilian population aged ≥18 which allows for state-level prevalence estimates for both diabetes and depression [16]. The survey also collects information data on demographic characteristics.

The BRFSS uses disproportionate stratified sampling for landline calls and random sampling of cell phones. When accounting for the survey's sampling and weighting, the BRFSS data are meant to be representative of the adult population residing in all 50 states, including the District of Columbia. The survey design was specified as follows, using STATA 15: svyset [pweight = _LLCPWT], strata(_STSTR) psu(_PSU) [17].

### Subjects

The 2016–2017 cohort comprises 477,665 respondents across 50 states and the District of Columbia. For 9% of this cohort, we do not have complete information on the outcomes and modifiers of interest. The final sample used in the analysis comprises 436,744 responders.

The following question identified individuals with depression in BRFSS: *Has a doctor, nurse, or other health professionals ever told you that you have a depressive disorder, including depression, major depression, dysthymia, or minor depression*?

The following question identified individuals with diabetes in BRFSS: *Has a doctor, nurse, or other health professionals ever told you that you have diabetes*? Women who were told they had diabetes only during pregnancy were excluded from the sample.

### Statistical approach

Depression and diabetes are both conditions that exist within a continuum and for which a diagnosis represents a dichotomous clinical adjudication, based on a discretizing threshold derived from an underlying latent variable. In the case of diabetes, the underlying latent variable would be fasting plasma glucose levels or glycated hemoglobin percentages and in the case of depression, it would be number of days a person experience loss of enjoyment, feelings of hopelessness or worthlessness and other related symptoms. The idea of a disease continuum underlying diagnosis informs the choice to model diabetes and depression as a bivariate normal distribution. All regression analyses control for state, sex, age, race, marital status, education, veteran status and lifestyle (exercise and smoking).

Each condition can be defined as a latent variable:

$$y_1^* = \beta_{y1} X + \epsilon_1, \quad y_1 = 1 \ (y_1^* > 0) \qquad \text{Eq 1}$$

$$y_2^* = \beta_{y2} X + \epsilon_2, \quad y_2 = 1 \ (y_2^* > 0) \qquad \text{Eq 2}$$

The goal is to estimate the coefficients needed to account for this joint distribution.
The latent correlation between diabetes ($y_1$) and depression ($y_2$) can be defined as:

$$\begin{matrix} \epsilon_1 \\ \epsilon_2 \end{matrix} \sim N \left[ \begin{matrix} 0 \\ 0 \end{matrix}, \begin{pmatrix} 1 & \rho \\ \rho & 1 \end{pmatrix} \right]$$

Where $\rho$ is the tetrachoric correlation between $y_1$ and $y_2$. It can be interpreted here as the correlation between the underlying diagnostic factors (before the application of thresholds) for diabetes and depression.

## Results and discussion

The tetrachoric correlation ($\rho$) between diabetes and depression is 17.2% and statistically significant. Older adults are more likely to be diagnosed with diabetes, but after the age of 65, they are less likely to be diagnosed with depression. Males are more likely to be diagnosed diabetes but less likely to be diagnosed depression. All races and ethnicities other than white

Caucasians are more likely to be diagnosed with diabetes and less likely to be diagnosed with depression. Native Americans have a higher prevalence for the co-occurrence of both conditions, compared to all other groups. Marital status, veteran status, school attainment, and exercise are the most important factors linked to the probability of having both diabetes and depression and operate in the same direction for both diabetes and depression (Table 1).

**Table 1. Bivariate diabetes and depression model coefficients.**

| Variables | diabetes | | | | | depression | | | | |
|---|---|---|---|---|---|---|---|---|---|---|
| | Coef. | SE | P>\|t\| | 95% CI (Lower, Upper) | | Coef. | SE | P>\|t\| | 95% CI (Lower, Upper) | |
| Male | 4.6% | 0.012 | < .0001 | 2.2% | 7.0% | -45.4% | 0.010 | < .0001 | -47.5% | -43.4% |
| non-Hispanic white | (omitted) | | | | | | | | | |
| non-Hispanic black | 26.2% | 0.017 | < .0001 | 22.8% | 29.7% | -31.5% | 0.018 | < .0001 | -35.0% | -28.0% |
| Hispanic | 17.3% | 0.021 | < .0001 | 13.3% | 21.4% | -35.2% | 0.018 | < .0001 | -38.8% | -31.7% |
| Asian | 33.3% | 0.045 | < .0001 | 24.5% | 42.0% | -48.7% | 0.044 | < .0001 | -57.3% | -40.0% |
| Native | 35.1% | 0.036 | < .0001 | 27.9% | 42.2% | 0.0% | 0.033 | 0.949 | -6.4% | 6.5% |
| Other | 11.9% | 0.034 | < .0001 | 5.0% | 18.8% | 8.1% | 0.031 | 0.007 | 2.1% | 14.2% |
| Age 18 to 24 | (omitted) | | | | | (omitted) | | | | |
| Age 25 to 29 | 12.7% | 0.051 | 0.009 | 2.8% | 22.6% | 0.1% | 0.024 | 0.808 | -4.7% | 4.8% |
| Age 30 to 34 | 38.7% | 0.048 | < .0001 | 29.3% | 48.1% | 0.0% | 0.024 | 0.768 | -4.8% | 4.8% |
| Age 35 to 39 | 62.6% | 0.045 | < .0001 | 53.8% | 71.4% | -0.2% | 0.025 | 0.768 | -5.2% | 4.8% |
| Age 40 to 44 | 84.1% | 0.044 | < .0001 | 75.6% | 92.6% | 0.8% | 0.026 | 0.516 | -4.4% | 6.0% |
| Age 45 to 49 | 104.4% | 0.042 | < .0001 | 96.1% | 112.7% | 3.2% | 0.026 | 0.089 | -1.9% | 8.2% |
| Age 50 to 54 | 119.9% | 0.041 | < .0001 | 112.0% | 127.8% | 0.6% | 0.024 | 0.452 | -4.1% | 5.4% |
| Age 55 to 59 | 137.1% | 0.041 | < .0001 | 129.1% | 145.1% | 3.4% | 0.024 | 0.059 | -1.4% | 8.1% |
| Age 60 to 64 | 150.3% | 0.041 | < .0001 | 142.5% | 158.2% | 2.7% | 0.025 | 0.110 | -2.2% | 7.5% |
| Age 65 to 69 | 165.5% | 0.041 | < .0001 | 157.6% | 173.4% | -9.6% | 0.025 | 0.001 | -14.6% | -4.6% |
| Age 70 to 74 | 170.4% | 0.042 | < .0001 | 162.3% | 178.5% | -21.9% | 0.027 | < .0001 | -27.1% | -16.6% |
| Age 75 to 79 | 167.8% | 0.044 | < .0001 | 159.3% | 176.3% | -38.5% | 0.032 | < .0001 | -44.8% | -32.2% |
| Age 80 or older | 157.1% | 0.044 | < .0001 | 148.6% | 165.6% | -52.6% | 0.032 | < .0001 | -58.8% | -46.4% |
| Veteran | 7.0% | 0.016 | < .0001 | 3.8% | 10.2% | 14.7% | 0.016 | < .0001 | 11.6% | 17.8% |
| BMI≥30 | 44.9% | 0.000 | < .0001 | 42.3% | 47.4% | 15.1% | 0.000 | < .0001 | 13.2% | 17.1% |
| Exercise | -24.8% | 0.012 | < .0001 | -27.2% | -22.4% | -22.7% | 0.011 | < .0001 | -24.8% | -20.5% |
| Daily smoker | 2.7% | 0.019 | 0.089 | -1.0% | 6.3% | 52.0% | 0.014 | < .0001 | 49.2% | 54.7% |
| Occasional smoker | -8.4% | 0.027 | 0.004 | -13.8% | -2.9% | 43.3% | 0.020 | < .0001 | 39.4% | 47.2% |
| Former smoker | 6.7% | 0.012 | < .0001 | 4.3% | 9.2% | 27.7% | 0.012 | < .0001 | 25.4% | 30.0% |
| Never smoker | (omitted) | | | | | (omitted) | | | | |
| Married | (omitted) | | | | | (omitted) | | | | |
| Divorced | 6.2% | 0.017 | < .0001 | 3.0% | 9.4% | 34.5% | 0.014 | < .0001 | 31.8% | 37.1% |
| Widowed | 10.3% | 0.019 | < .0001 | 6.7% | 13.9% | 21.8% | 0.019 | < .0001 | 18.2% | 25.5% |
| Separated | 9.2% | 0.034 | 0.006 | 2.6% | 15.8% | 43.0% | 0.028 | < .0001 | 37.5% | 48.5% |
| Never married | 11.0% | 0.019 | < .0001 | 7.3% | 14.7% | 27.2% | 0.015 | < .0001 | 24.3% | 30.1% |
| Member of an Unmarried couple | -0.2% | 0.038 | 0.732 | -7.5% | 7.2% | 23.6% | 0.024 | < .0001 | 18.8% | 28.3% |
| Did not graduate from high school | 37.8% | 0.020 | < .0001 | 33.9% | 41.7% | 20.3% | 0.018 | < .0001 | 16.8% | 23.9% |
| Graduated high school | 20.5% | 0.014 | < .0001 | 17.8% | 23.3% | 0.9% | 0.013 | 0.387 | -1.6% | 3.3% |
| Some college | 18.1% | 0.014 | < .0001 | 15.2% | 20.9% | 8.3% | 0.012 | < .0001 | 6.0% | 10.6% |
| Graduated college | (omitted) | | | | | (omitted) | | | | |
| Constant | -2.85 | 0.047 | < .0001 | -2.94 | -2.77 | -0.93 | 0.028 | < .0001 | -0.98 | -0.88 |
| ρ | 17.2% | 0.008 | < .0001 | 0.157 | 0.188 | | | | | |

(omitted) flags the references categories have been omitted. ρ is the tetrachoric correlation between diabetes and depression.

The marginal effects reported in Table 2 are based on the joint probability of diabetes and depression. Because we have two outcomes of interest, overall, we have four joint probabilities. Table 2 reports three out of four joint probabilities and not **P(y1 = 0, y2 = 0)** because the marginal effects, across the four joint probabilities, sum up to zero. Probabilities have been computed holding covariates at the population mean. Table 2 shows that most demographic characteristics, sex, age, and race operate in opposite directions in predicting diabetes and depression. Behavioral indicators, i.e., BMI, smoking, and exercise; and life outcomes, i.e., schooling, marital and veteran status, on the other hand, work in the same direction to produce co-occurrence.

**Table 2. Marginal effects for the joint probabilities.**

| Variables | diabetes and depression P(y1 = 1, y2 = 1) | | | | diabetes only P(y1 = 1, y2 = 0) | | | | depression only P(y1 = 0, y2 = 1) | | | |
|---|---|---|---|---|---|---|---|---|---|---|---|---|
| | dy/dx | P>\|t\| | [95% Conf. Interval] | | dy/dx | P>\|t\| | [95% Conf. Interval] | | dy/dx | P>\|t\| | [95% Conf. Interval] | |
| Male | -0.79% | < .0001 | -0.87% | -0.70% | 1.37% | < .0001 | 1.1% | 1.6% | -9.78% | < .0001 | -10.21% | -9.35% |
| non-Hispanic black | -0.02% | 0.777 | -0.14% | 0.10% | 3.88% | < .0001 | 3.4% | 4.4% | -6.45% | < .0001 | -6.99% | -5.92% |
| Hispanic | -0.29% | < .0001 | -0.41% | -0.18% | 2.70% | < .0001 | 2.2% | 3.2% | -6.95% | < .0001 | -7.50% | -6.40% |
| Asian | -0.28% | 0.038 | -0.54% | -0.02% | 5.53% | < .0001 | 4.0% | 7.0% | -8.65% | < .0001 | -9.60% | -7.70% |
| Native | 1.17% | < .0001 | 0.79% | 1.56% | 4.53% | < .0001 | 3.4% | 5.7% | -1.17% | 0.074 | -2.45% | 0.12% |
| Other | 0.56% | 0.001 | 0.24% | 0.87% | 1.09% | 0.006 | 0.3% | 1.9% | 1.43% | 0.038 | 0.08% | 2.78% |
| Age 25 to 29 | 0.37% | 0.028 | 0.04% | 0.70% | 1.37% | 0.020 | 0.2% | 2.5% | -0.35% | 0.503 | -1.39% | 0.68% |
| Age 30 to 34 | 1.28% | < .0001 | 0.87% | 1.69% | 4.91% | < .0001 | 3.5% | 6.4% | -1.27% | 0.013 | -2.27% | -0.26% |
| Age 35 to 39 | 2.29% | < .0001 | 1.82% | 2.76% | 9.21% | < .0001 | 7.5% | 10.9% | -2.34% | < .0001 | -3.34% | -1.34% |
| Age 40 to 44 | 3.40% | < .0001 | 2.84% | 3.97% | 13.84% | < .0001 | 11.9% | 15.8% | -3.22% | < .0001 | -4.20% | -2.24% |
| Age 45 to 49 | 4.72% | < .0001 | 4.08% | 5.36% | 18.88% | < .0001 | 16.8% | 21.0% | -3.96% | < .0001 | -4.88% | -3.04% |
| Age 50 to 54 | 5.33% | < .0001 | 4.71% | 5.96% | 23.01% | < .0001 | 20.9% | 25.1% | -5.18% | < .0001 | -6.00% | -4.37% |
| Age 55 to 59 | 6.65% | < .0001 | 5.95% | 7.35% | 28.04% | < .0001 | 25.7% | 30.4% | -5.85% | < .0001 | -6.65% | -5.05% |
| Age 60 to 64 | 7.36% | < .0001 | 6.63% | 8.09% | 32.11% | < .0001 | 29.8% | 34.4% | -6.72% | < .0001 | -7.48% | -5.97% |
| Age 65 to 69 | 6.83% | < .0001 | 6.11% | 7.56% | 39.11% | < .0001 | 36.7% | 41.6% | -8.99% | < .0001 | -9.60% | -8.39% |
| Age 70 to 74 | 5.69% | < .0001 | 5.02% | 6.36% | 42.94% | < .0001 | 40.3% | 45.6% | -10.29% | < .0001 | -10.83% | -9.76% |
| Age 75 to 79 | 3.91% | < .0001 | 3.26% | 4.56% | 44.27% | < .0001 | 41.4% | 47.1% | -11.33% | < .0001 | -11.80% | -10.85% |
| Age 80 or older | 2.50% | < .0001 | 2.00% | 2.99% | 41.36% | < .0001 | 38.5% | 44.3% | -11.90% | < .0001 | -12.34% | -11.46% |
| Veteran | 0.54% | < .0001 | 0.40% | 0.68% | 0.38% | 0.024 | 0.1% | 0.7% | 3.11% | < .0001 | 2.37% | 3.85% |
| BMI≥30 | 1.44% | < .0001 | 1.35% | 1.52% | 3.94% | < .0001 | 3.7% | 4.2% | 2.05% | < .0001 | 1.65% | 2.46% |
| Exercise | -1.35% | < .0001 | -1.46% | -1.23% | -2.12% | < .0001 | -2.4% | -1.8% | -4.28% | < .0001 | -4.78% | -3.79% |
| Daily smoker | 1.31% | < .0001 | 1.12% | 1.50% | -0.96% | < .0001 | -1.3% | -0.6% | 13.33% | < .0001 | 12.51% | 14.15% |
| Occasional smoker | 0.64% | < .0001 | 0.40% | 0.87% | -1.64% | < .0001 | -2.0% | -1.2% | 11.48% | < .0001 | 10.31% | 12.66% |
| Former smoker | 0.82% | < .0001 | 0.71% | 0.92% | 0.06% | 0.630 | -0.2% | 0.3% | 6.13% | < .0001 | 5.57% | 6.69% |
| Divorced | 1.01% | < .0001 | 0.85% | 1.17% | -0.19% | 0.219 | -0.5% | 0.1% | 8.17% | < .0001 | 7.45% | 8.90% |
| Widowed | 0.85% | < .0001 | 0.67% | 1.03% | 0.55% | 0.006 | 0.2% | 0.9% | 4.75% | < .0001 | 3.83% | 5.68% |
| Separated | 1.42% | < .0001 | 1.03% | 1.81% | -0.17% | 0.580 | -0.8% | 0.4% | 10.71% | < .0001 | 9.08% | 12.33% |
| Never married | 0.96% | < .0001 | 0.80% | 1.11% | 0.49% | 0.011 | 0.1% | 0.9% | 5.88% | < .0001 | 5.17% | 6.59% |
| Member of an Unmarried couple | 0.51% | 0.001 | 0.21% | 0.80% | -0.53% | 0.121 | -1.2% | 0.1% | 5.61% | < .0001 | 4.36% | 6.87% |
| Did not graduate from high school | 1.93% | < .0001 | 1.69% | 2.17% | 3.95% | < .0001 | 3.4% | 4.5% | 3.19% | < .0001 | 2.34% | 4.03% |
| Graduated high school | 0.61% | < .0001 | 0.50% | 0.72% | 2.18% | < .0001 | 1.9% | 2.5% | -0.41% | 0.124 | -0.93% | 0.11% |
| Some college | 0.71% | < .0001 | 0.59% | 0.82% | 1.70% | < .0001 | 1.4% | 2.0% | 1.27% | < .0001 | 0.76% | 1.78% |

The derivative of y with respect to x (dy/dx) represent the marginal probabilities. *Exercise* is defined as a binary response to the question: During the past month, other than your regular job, did you participate in any physical activities or exercises such as running, calisthenics, golf, gardening, or walking for exercise?

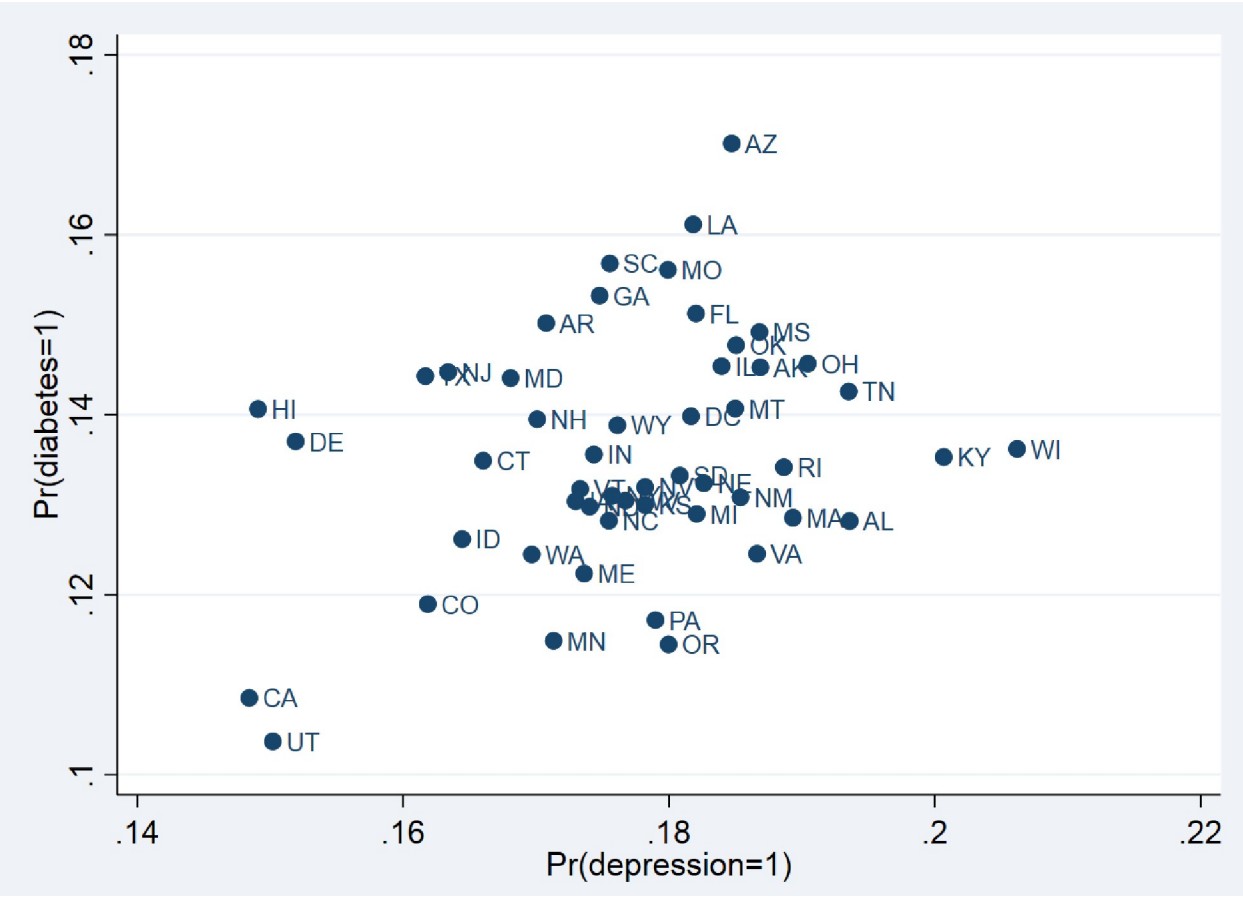

**Fig 1. Marginal probabilities of diabetes and depression across states.** The vertical axis shows the state's average marginal probability of being diagnosed with diabetes and the horizontal axis shows the state's average marginal probability of being diagnosed with depression.

Fig 1 shows that depression alone is more prevalent than diabetes across all states. The vertical and horizontal axes show the average marginal probability across states of having either a diabetes or depression diagnosis, respectively. Thirteen states (Alaska, Arizona, D.C., Florida, Illinois, Louisiana, Missouri, Mississippi, Montana, Ohio, Oklahoma, Tennessee, and Wisconsin) have a prevalence of diabetes and depression exceeding the national *marginal* averages (13.5% for diabetes and 17.6% for depression are slightly higher than the national prevalence for each condition). Within these states, the probability of any one person being diagnosed with both conditions **P(y1 = 1, y2 = 1)** is as high as 4.3% (e.g., Arizona). Ten states (Arkansas, Delaware, Georgia, Hawaii, Maryland, New Hampshire, New Jersey, South Carolina, Texas, and Wyoming) have a higher prevalence of diabetes than the national average but also a reported diagnosed depression prevalence below the national average. Thirteen states (Alabama, Kansas, Kentucky, Massachusetts, Michigan, Nebraska, New Mexico, Nevada, Oregon, Pennsylvania. Rhode Island, South Dakota, and Virginia) have a higher prevalence of depression than the national average but also a prevalence of diabetes below the national average. The remaining 15 states have a lower reported prevalence of diabetes and depression than the national average. Within these states, the probability of any one person being diagnosed with both conditions (joint probability) is as low as 2.3% (e.g., Utah).

The vertical and horizontal lines in Fig 2 show the unconditional mean of diabetes and depression (10.7% and 16.6%, respectively) as well as the 0.5 probability threshold for each

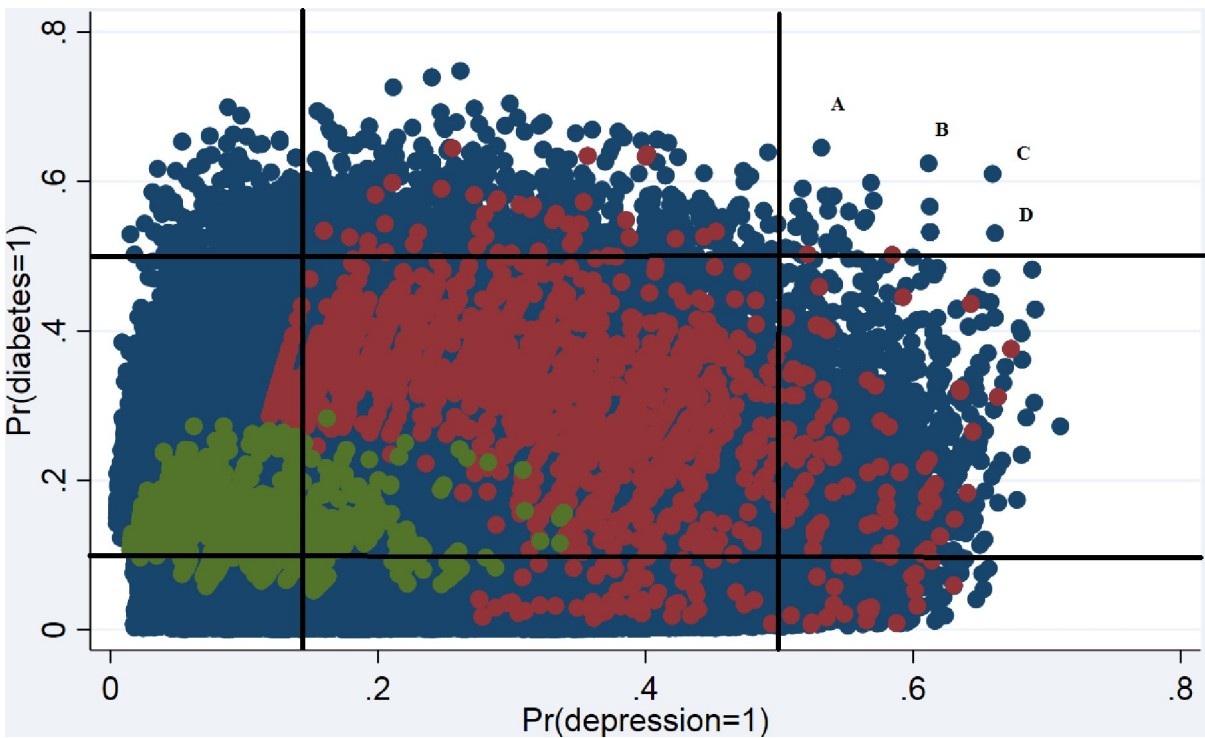

**Fig 2. Predicting probabilities of diabetes and depression at the patient level.** The vertical axis and horizontal axes shows the patient's probability of being diagnosed with diabetes and depression, respectively. Green dots represent non-veterans, non-white females, with a BMI<30, never smoked, currently married, exercise, and finished college. Red dots represent white men, veterans, with a BMI≥30, smokers, unmarried, and without a college degree.

condition (black lines). Point A (coordinates 0.661, 0.531) represents a Native American woman between the ages of 65 and 69, with BMI≥30, an occasional smoker, who did not exercise in the 30 days prior to being surveyed, did not finish high school and is divorced. Point B (coordinates) represents a Native American woman between the ages of 55 and 59, with BMI≥30, a daily smoker, who did not exercise in the past 30 days, did not finish high school, and is divorced. Point C (coordinates 0.610, 0.659) represents a Native American woman between the ages of 55 and 59, with BMI≥30, a daily smoker, who did not exercise in the past 30 days, did not finish high school, widowed. Point D (coordinates 0.532, 0.613) represents a woman of a non-pre-specified race, between the ages of 60 and 64, with BMI≥30, daily smoker, who did not exercise, did not finish high school, and is widowed. To highlight the fact that behavioral variables are more important than demographic variables, Fig 2 shows in green individuals that have a riskier demographic profile (non-white females) compared to white males but that at the same time have the most protective socio-economic indicators, i.e., they are non-veterans, with a BMI<30, never smoked, currently married, exercise, and finished college. Highlighted in red are the opposite: white men with riskier socio-economic profiles—veterans, with a BMI≥30, that smoke occasionally or daily, that are unmarried, that do not exercise and that did not finish college. The average person in the sample with BMI<30, never smoked, currently married, exercise, and finished college has 1.3% (CI:1.1%- 1.5%) probability of having both diabetes and depression, while an otherwise comparable person with the opposite lifestyle has a probability of 16.1% (CI: 14.6%- 17.6%).

## Conclusions

While the probability of a single condition, irrespective of the other (i.e. the marginal probability) is relatively high for both conditions, the probability of being diagnosed with both conditions is currently very low, approximately 3%. This is problematic because the latent correlation of diabetes and depression is at least 5 times higher (17.2%). The reason for this discrepancy could be due to a systematic under-diagnosed of particularly vulnerable and at-risk individuals. It is difficult to combat diseases when the number of underdiagnosed cases is high. Estimates report that around two-thirds of all cases of depression are undiagnosed [16] and that approximately one-third of all cases of diabetes in the U.S. are undiagnosed [18].

The U.S. Preventive Services Task Force (USPSTF) now recommends screening for depression in the general adult population and notes that persons with chronic illnesses, such as diabetes, are at increased risk of developing depression. The USPSTF recommends screening should have adequate systems in place to ensure accurate diagnosis, effective treatment, and appropriate follow-up, albeit it does not consider the costs of providing this service in its assessment [19]. Effective care for depression in patients with diabetes is challenging. Mental health services are often separated from primary care leading to poor access to psychological services and other effective treatments. Primary care is an important entry point for diagnosis but primary care providers' time is also limited. Harrison et al. (2010) found that females have higher odds of being screened for depression compared to males and that individuals ≥ 65 years of age are significantly less likely to be screened for depression than 40–64 years olds [20]. There is also evidence to suggest that depression is underdiagnosed in minority ethnic groups, including African Americans and Hispanics (Shao et al, 2016). Lack of reimbursement incentives or missing information in patient's medical records may also play an important role in healthcare providers' screening behavior [21]. Underdiagnoses may lead to a low estimated prevalence of depression in certain groups. Low screening rates translate into missed opportunities for treatment. More research is therefore needed to understand the determinants of undiagnosed cases for both diabetes and depression to avoid algorithmic bias.

Given the fragmentation of care, limited funding, high costs of co-occurrence to the individual and the healthcare system, and that the reported co-occurrence is relatively small, having a clinical instrument for rapid and efficient diagnoses, that will help us deploy scarce resources to those most in need seems paramount. Leading questions about lifestyle choices, educational attainment and family life are simple and likely effective strategies to screen patients with diabetes and at risk for depression. Asking targeted questions might also have spillover benefits beyond the diagnosis of co-occurrence by increasing patients' satisfaction.

At the state level, quality improvement initiatives may be more successful if targeted to areas where the prevalence of both diabetes and mental health issues are high and where there are limited resources dedicated to mental health services.

## Author Contributions

**Conceptualization:** Maria L. Alva.

**Data curation:** Maria L. Alva.

**Formal analysis:** Maria L. Alva.

**Funding acquisition:** Maria L. Alva.

**Investigation:** Maria L. Alva.

**Methodology:** Maria L. Alva.

**Project administration:** Maria L. Alva.

**Resources:** Maria L. Alva.

**Software:** Maria L. Alva.

**Supervision:** Maria L. Alva.

**Validation:** Maria L. Alva.

**Visualization:** Maria L. Alva.

**Writing – original draft:** Maria L. Alva.

**Writing – review & editing:** Maria L. Alva.

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
