## [Decision Letter · Decision Letter 0]

15 Jan 2020

PONE-D-19-22449

Co-occurrence of diabetes and depression in the U.S.: How can we target those at risk better?

PLOS ONE

Dear Dr Alva,  

Thank you for submitting your manuscript to PLOS ONE. After careful consideration, we feel that it has merit but does not fully meet PLOS ONE’s publication criteria as it currently stands. Therefore, we invite you to submit a revised version of the manuscript that addresses the points raised during the review process.

We would appreciate receiving your revised manuscript by Feb 29 2020 11:59PM. To enhance the reproducibility of your results, we recommend that if applicable you deposit your laboratory protocols in protocols.io, where a protocol can be assigned its own identifier (DOI) such that it can be cited independently in the future. For instructions see: http://journals.plos.org/plosone/s/submission-guidelines#loc-laboratory-protocols

We look forward to receiving your revised manuscript.

Kind regards,

Abebaw Fekadu, MD/PhD

Academic Editor

PLOS ONE

Journal Requirements:

Bogner, Hillary R., and Heather F. de Vries McClintock. "Costs of coexisting depression and diabetes." (2016): 594-595.

In your revision ensure you cite all your sources (including your own works), and quote or rephrase any duplicated text outside the methods section. Further consideration is dependent on these concerns being addressed.

4. Thank you for stating the following in the Financial Disclosure section:"The author received no specific funding for this work."

Thank you for stating the following in the Competing Interests section:"The author declares that no competing interests exist."

We note that one or more of the authors are employed by a commercial company: "IMPAQ International,"

a)Please provide an amended Funding Statement declaring this commercial affiliation, as well as a statement regarding the Role of Funders in your study. If the funding organization did not play a role in the study design, data collection and analysis, decision to publish, or preparation of the manuscript and only provided financial support in the form of authors' salaries and/or research materials, please review your statements relating to the author contributions, and ensure you have specifically and accurately indicated the role(s) that these authors had in your study. You can update author roles in the Author Contributions section of the online submission form.

b). Please also provide an updated Competing Interests Statement declaring this commercial affiliation along with any other relevant declarations relating to employment, consultancy, patents, products in development, or marketed products, etc. 

Additional Editor Comments:

Title:

I suggest modifying title. The “risk” factors described are weakly indicative of “co-occurrence” and recommendations provided are not robust enough to include 'risk' within the title.

Abstract:

Please provide full name for BRFSS

Background:

Good background overall.

Can make more a bit more detail in the background on existing studies of state level variation in the distribution of the conditions of interest and what the current report may add.

Design: The description in this section is one of statistical analytic approach rather than design. This needs rewriting.

Results:

Correlation of 0.17 is very low despite statistical significance. This needs to be reflected upon.

All “exhibit” should be relabelled as Figure.

Table: Would be helpful for readers to be explicit about what the coefficients are representing.

In the text description of “Exhibit 3” (page 7, line 21-22), the detailed reference to the axis should go as an explanatory note under the exhibit itself.

Discussion:

Does not pay attention in any depth to the state level differences and why comorbidity at the state level may be explained.

It would have also been good to have concluding remarks or reflections at the end.

Reviewers' comments:

Reviewer's Responses to Questions

**Comments to the Author**

1. Is the manuscript technically sound, and do the data support the conclusions?

Reviewer #1: Yes

Reviewer #2: Partly

2. Has the statistical analysis been performed appropriately and rigorously? 

Reviewer #1: I Don't Know

Reviewer #2: No

3. Have the authors made all data underlying the findings in their manuscript fully available?

Reviewer #1: Yes

Reviewer #2: Yes

4. Is the manuscript presented in an intelligible fashion and written in standard English?

Reviewer #1: Yes

Reviewer #2: Yes

5. Review Comments to the Author

Reviewer #1: I would like more information about why the authors used the data analysis they chose. I also feel that the one scatter plot on individuals could be re-done . it is impossible to decipher. I also feel the conclusion is weak and could be re-done to highlight why the article is important

Reviewer #2: The manuscript entitled 'Co-occurrence of diabetes and depression in the U.S.: How can we target those at risk better?' with the aim to evaluate demographic and socio-economic indicators associated with both depression and diabetes at the macro (across-states) and micro level (individuals)'

This is quite an interesting study. However, the manuscript requires improvement based on the following comments.

Abstracts

Page 2 Line 12, BMI >30 or BMI≥30? Likewise in discussion section i.e. Page 8 Line 17, 19, Page 9 Line 1.

Try to avoid using active sentence i.e 'I' (Page 5 Line 7,22; Page 6 Line 10)

Page 5, more information/description to be provided on BRFSS such as data type, type of measurement/ test/ inventories/ questionnaires etc i.e measure for diabetes, depression etc

Page 5 Line 10-13, more information to be provided.

Subjects

Page 5 Line 16-23, this section requires revision to state that these subjects (depression and/or diabetes) were identified in BRFSS via the two questions which were found in BRFSS.

Page 5 Line 16-17, to clearly state how many subjects were analyzed and to state if any missing data occurred in the database that was used in the analysis.

Design

Page 6 Line 6, 7, the words 'could be', 'might be' to be avoided.

Page 6 Line 13, 14, 16, 17, the equation to be labelled.

More information to be provided on the statistical analysis such as type of regression analysis, model fit, level of acceptance significance, 1 or 2 tailed test etc and description on how the analyses were performed on state and individual level. The statistical software STATA including the publisher name and version to be stated.

Results

Page 7 Line 2, 17% to be replaced with 17.2%.

The word Exhibit to be replaced with word Table.

Exhibit 1, technically p value cannot be written as 0.000 (to use symbol < to denote less). Denote what 'omitted' and 'ρ' refers to in the table footnote. CIL and CIU to be written as 95% CI (Lower, Upper). P>[t] to be written as P. Symbol >= to be replaced with symbol ≥. The category for BMI and veteran to be stated in the table footnote.

Likewise with Exhibit 2. dy/dx to be denoted in table footnote.

Page 7 Line 22 - Page 8 Line 11, total states added up to 51. On Page 5 Line 16, it was stated 50.

Page 9 Line 10 and 12, CI to be written as 95%CI.

Overall, the writeup requires revision in terms of flow, clarity and more information to be provided.

6. PLOS authors have the option to publish the peer review history of their article (what does this mean?). If published, this will include your full peer review and any attached files.

Reviewer #1: No

Reviewer #2: No

---

## [Author Response · Author response to Decision Letter 0]

17 Feb 2020

Editor’s comments

Answer. Thank you. The title page now follows PLOS ONE style template. My affiliation has also been updated. In the main text – all headings follow the names used in the guideline, i.e., I now call “introduction” what was previously defined as “background”.

Bogner, Hillary R., and Heather F. de Vries McClintock. "Costs of coexisting depression and diabetes." (2016): 594-595. In your revision ensure you cite all your sources (including your own works), and quote or rephrase any duplicated text outside the methods section. Further consideration is dependent on these concerns being addressed.

Answer. Any overlapping text is accidental. I used here the software that I use to evaluate plagiarism in my students’ essays. The only parallel I found is regarding this section: “The global burden of disease study … prevalence” I have deleted that sentence because it does not change in any meaningful way the arguments in the text and I have added instead other evidence from the literature to highlight the importance of the paper.

If you have specific text suggestions that have not been captured by software, please kindly point those to me as I am unable to identify them.

Answer. The dataset I have used in my analysis is publicly available. Reference 16 provides the links to access the data.

4. Thank you for stating the following in the Financial Disclosure section:"The author received no specific funding for this work."

Thank you for stating the following in the Competing Interests section:"The author declares that no competing interests exist."

We note that one or more of the authors are employed by a commercial company: "IMPAQ International,"

Answer. My former employer, IMPAQ international is not a commercial company but a (for profit) research organization. I was a senior economist there, and all researchers of that institution as well as similar ones have to undergo background checks that guarantee they do not have conflicts of interest. More information about my former employer is available here: https://www.impaqint.com/. While the paper was under review I have become a professor at Georgetown University.

a)Please provide an amended Funding Statement declaring this commercial affiliation, as well as a statement regarding the Role of Funders in your study. If the funding organization did not play a role in the study design, data collection and analysis, decision to publish, or preparation of the manuscript and only provided financial support in the form of authors' salaries and/or research materials, please review your statements relating to the author contributions, and ensure you have specifically and accurately indicated the role(s) that these authors had in your study. You can update author roles in the Author Contributions section of the online submission form.

Answer. All parts of the study (the study design, data collection and analysis, decision to publish, or preparation of the manuscript) were done on my own time and not during IMPAQ’s working hours. Therefore, IMPAQ was not the “funding organization” as my salary then reflected the hours spent on contract work and did not cover projects done in my own time. Also please note my change in affiliation to a university institution.

Answer. Both at IMPAQ (previously) and at Georgetown University (not) I (dis not) do not get a salary to write papers. I write papers on my own time and interest-- I have no conflict of interests with any organization that might draw any benefit from my independent research. I am happy to state the truth and to explain further my circumstances, with the amount of detail the editors suggest would be appropriate. However, the suggested statement does not reflect my situation.

Answer. As aforementioned, my former employer is not a commercial company but an interdependent research organization whereby employees have to be able to demonstrate no COIs. My current employer is a University, with the same requirements.

b). Please also provide an updated Competing Interests Statement declaring this commercial affiliation along with any other relevant declarations relating to employment, consultancy, patents, products in development, or marketed products, etc. 

Answer. I do not have now (or ever had) a commercial affiliation. I hope my previous answers clarify any confusion related to my former employer.

Answer. The data I used (BRFSS) in my analysis is publicly available. I have included a link in the paper (reference number 16) to where readers can download the data. 

Additional Editor Comments:

Title: I suggest modifying title. The “risk” factors described are weakly indicative of “co-occurrence” and recommendations provided are not robust enough to include 'risk' within the title are not robust enough to include 'risk' within the title.

Answer. As requested, I deleted from the title “How can we target those at risk better?”

Abstract: Please provide full name for BRFSS

Answer. Done. The text in the abstract now reads “Behavioral Risk Factor Surveillance System”

Background: Good background overall. Can make more a bit more detail in the background on existing studies of state level variation in the distribution of the conditions of interest and what the current report may add.

Answer. Thank you. More details in the background has been added on studies that state national level variation. There are no peer reviewed studies looking at differences across states.

Design: The description in this section is one of statistical analytic approach rather than design. This needs rewriting.

Answer. Following the editor’s advice, I now use the term “statistical approach” rather than design. 

Results: Correlation of 0.17 is very low despite statistical significance. This needs to be reflected upon. All “exhibit” should be relabelled as Figure. Table: Would be helpful for readers to be explicit about what the coefficients are representing.

Answer. Thank you. All exhibits have now been re-labeled as figures and tables.

In the text description of “Exhibit 3” (page 7, line 21-22), the detailed reference to the axis should go as an explanatory note under the exhibit itself.

Answer. Thank you. A note has now been added under Figure 1, as requested.

Discussion: Does not pay attention in any depth to the state level differences and why comorbidity at the state level may be explained. It would have also been good to have concluding remarks or reflections at the end.

Answer. Thank you. In the results and discussion, differences across states are explained in depth – however, understanding why these differences come about are out of the scope of this paper, as the data available does not allow us to answer that question.

Reviewers' comments:

Reviewer's Responses to Questions

Comments to the Author

1. Is the manuscript technically sound, and do the data support the conclusions?

Reviewer #1: Yes

Reviewer #2: Partly

2. Has the statistical analysis been performed appropriately and rigorously? 

Reviewer #1: I Don't Know

Reviewer #2: Yes

3. Have the authors made all data underlying the findings in their manuscript fully available?

Reviewer #1: Yes

Reviewer #2: Yes

4. Is the manuscript presented in an intelligible fashion and written in standard English?

Reviewer #1: Yes

Reviewer #2: Yes

5. Review Comments to the Author

Reviewer #1: I would like more information about why the authors used the data analysis they chose. I also feel that the one scatter plot on individuals could be re-done . it is impossible to decipher. I also feel the conclusion is weak and could be re-done to highlight why the article is important

Answer. Reviewer #1 does not provide any guidance on how s/he would like Figure 2 to be changed or why it is hard for s/he to understand it. I added footnoted to assist the reader with interpretation.

Reviewer #2: The manuscript entitled 'Co-occurrence of diabetes and depression in the U.S.: How can we target those at risk better?' with the aim to evaluate demographic and socio-economic indicators associated with both depression and diabetes at the macro (across-states) and micro level (individuals)'

This is quite an interesting study. However, the manuscript requires improvement based on the following comments.

Abstracts Page 2 Line 12, BMI >30 or BMI BMI≥3030? Likewise in discussion section i.e. Page 8 Line 17, 19, Page 9 Line 1.

Answer. Thank you. Obesity was denoted with BMI≥30. This has been amended in the text and made consistent with the computations.

Try to avoid using active sentence i.e 'I' (Page 5 Line 7,22; Page 6 Line 10)

Answer. All active sentences were avoided. On page 5, I now use “This analysis uses”. On page 6, line 2, I changed the previously active voice into passive voice: “ Women….were excluded from the sample”. On page 6, line 13, I now write “All analyses control for…”

Page 5, more information/description to be provided on BRFSS such as data type, type of measurement/ test/ inventories/ questionnaires etc 

i.e measure for diabetes, depression etc

Page 5 Line 10-13, more information to be provided.

Answer. I added clarification on the type of data available in BRFSS as requested

Subjects Page 5 Line 16-23, this section requires revision to state that these subjects (depression and/or diabetes) were identified in BRFSS via the two questions which were found in BRFSS.

Answer. The fact that the questions are found in BRFSS is now stated in the text as requested

Page 5 Line 16-17, to clearly state how many subjects were analyzed and to state if any missing data occurred in the database that was used in the analysis.

Answer. The information requested is stated on page 5, line 21. 477,665 respondents answered the question about diabetes and depression. 40,921.00 (or 9% of the sample) did not have complete information on all the demographic characteristics the study included and therefore were excluded from the sample. I re-edited the section to make the exposition clearer.

Design Page 6 Line 6, 7, the words 'could be', 'might be' to be avoided.

Answer. “Could be” and “might be” gave been substituted with “would”

Page 6 Line 13, 14, 16, 17, the equation to be labelled.

Answer. The equations (previously in line 13 and 14) were label as the reviewer suggested. The correlation matrix was not labelled because that is NOT an equation not it representants two different arguments like the reviewer seems to indicate. It is one single argument.

More information to be provided on the statistical analysis such as type of regression analysis, model fit, level of acceptance significance, 1 or 2 tailed test etc and description on how the analyses were performed on state and individual level. The statistical software STATA including the publisher name and version to be stated.

Answer. On page 5, it was already stated that the statistical software used is STATA 15. It is also stated that I use a bivariate normal model. Standard errors, P-values and confidence intervals are presented in Tables 1 and 2. The study is descriptive and does not perform differences across cohorts therefore the comment being made on tests is not relevant in this case.

Results Page 7 Line 2, 17% to be replaced with 17.2%.

Answer. The change requested was done.

The word Exhibit to be replaced with word Table.

Answer. The change requested was done.

Exhibit 1, technically p value cannot be written as 0.000 (to use symbol < to denote less). Denote what 'omitted' and 'ρ' refers to in the table footnote. CIL and CIU to be written as 95% CI (Lower, Upper). P>[t] to be written as P. Symbol >= to be replaced with symbol ≥. The category for BMI and veteran to be stated in the table footnote. Likewise with Exhibit 2. dy/dx to be denoted in table footnote.

Answer. 0.000 was replaced through with <0.0001 as requested. A footnote has been added under table 1, as requested. 95% CI (Lower, Upper) has been written in lieu of CIL CIU. The Symbol >= has been replace.

A footnote for dy/dx has been added in the footnote under Exhibit 2 – now labelled as Table 2. Note however that the table already explains that dy/dx means: “marginal probability”.

Page 7 Line 22 - Page 8 Line 11, total states added up to 51. On Page 5 Line 16, it was stated 50.

Answer. It is 50 states plus DC. The counts add up to 51 and are consistent with the description.

Page 9 Line 10 and 12, CI to be written as 95%CI.

Answer. The change requested was added.

---

## [Decision Letter · Decision Letter 1]

18 May 2020

PONE-D-19-22449R1

Co-occurrence of diabetes and depression in the U.S.

PLOS ONE

Dear Dr. Alva,

Thank you for submitting your manuscript to PLOS ONE. After careful consideration, we feel that it has merit but does not fully meet PLOS ONE’s publication criteria as it currently stands. Therefore, we invite you to submit a revised version of the manuscript that addresses the points raised during the review process.

Please make minor revisions following the reviewers' comments.

We would appreciate receiving your revised manuscript by Jul 02 2020 11:59PM. To enhance the reproducibility of your results, we recommend that if applicable you deposit your laboratory protocols in protocols.io, where a protocol can be assigned its own identifier (DOI) such that it can be cited independently in the future. For instructions see: http://journals.plos.org/plosone/s/submission-guidelines#loc-laboratory-protocols

We look forward to receiving your revised manuscript.

Kind regards,

Nayu Ikeda, Ph.D.

Academic Editor

PLOS ONE

Reviewers' comments:

Reviewer's Responses to Questions

**Comments to the Author**

1. If the authors have adequately addressed your comments raised in a previous round of review and you feel that this manuscript is now acceptable for publication, you may indicate that here to bypass the “Comments to the Author” section, enter your conflict of interest statement in the “Confidential to Editor” section, and submit your "Accept" recommendation.

Reviewer #1: All comments have been addressed

Reviewer #2: (No Response)

2. Is the manuscript technically sound, and do the data support the conclusions?

Reviewer #1: Yes

Reviewer #2: Partly

3. Has the statistical analysis been performed appropriately and rigorously? 

Reviewer #1: I Don't Know

Reviewer #2: (No Response)

4. Have the authors made all data underlying the findings in their manuscript fully available?

Reviewer #1: Yes

Reviewer #2: Yes

5. Is the manuscript presented in an intelligible fashion and written in standard English?

Reviewer #1: Yes

Reviewer #2: Yes

6. Review Comments to the Author

Reviewer #1: I was concerned on the previous review that the graphs were not explained in any way. The author has added text below the graphs to explain what they show. I would remind the author to make the manuscript reader friendly, as everyone who reads this may not be in the field of the writer.

Reviewer #2: Minor comments

Table 1, for Did not graduate from high school, the figures 33.9% 41.7% , 16.8% 23.9% to be aligned parallel with other figures.

Table 2, Diabetes only P(y1=1, y2=0) (Non hispanic black, hispanic, native, others) typo <.0001<.0001.

The word marginal probability in the first column to be replaced with variables.

STATA which was stated for 'The survey design was specified as follows, using Stata 15: svyset [pweight=_LLCPWT], strata(_STSTR) psu(_PSU) to be stated in statistical analysis section with citation 'STATA: Release 15. College Station, TX: StataCorp LLC' and the level of accepted statistical significant.

Some references did not conform to the journal format.

7. PLOS authors have the option to publish the peer review history of their article (what does this mean?). If published, this will include your full peer review and any attached files.

Reviewer #1: No

Reviewer #2: No

---

## [Author Response · Author response to Decision Letter 1]

18 May 2020

Thank you for the comments. These have been fully addressed both the marked and unmarked copies uploaded.

My response to each suggestion raised is reported below:

• Table 1, for Did not graduate from high school, the figures 33.9% 41.7% , 16.8% 23.9% to be aligned parallel with other figures.

Answer: Thank you. The table was re-formatted

• Table 2, Diabetes only P(y1=1, y2=0) (Non hispanic black, hispanic, native, others) typo <.0001<.0001.

Answer: Thank you. The typo has now been corrected.

• The word marginal probability in the first column to be replaced with variables.

Answer: Thank you. This was changed.

• STATA which was stated for 'The survey design was specified as follows, using Stata 15: svyset [pweight=_LLCPWT], strata(_STSTR) psu(_PSU) to be stated in statistical analysis section with citation 'STATA: Release 15. College Station, TX: StataCorp LLC' and the level of accepted statistical significant.

Answer: The STATA citation has been added and all the references have been reformatted.

• Some references did not conform to the journal format.

Answer: All references have now been formatted using the ICMJE style

---

## [Editor Report · Decision Letter 2]

2 Jun 2020

Co-occurrence of diabetes and depression in the U.S.

PONE-D-19-22449R2

Dear Dr. Alva,

We’re pleased to inform you that your manuscript has been judged scientifically suitable for publication and will be formally accepted for publication once it meets all outstanding technical requirements.

Kind regards,

Nayu Ikeda, Ph.D.

Academic Editor

PLOS ONE
---

## [Editor Report · Acceptance letter]

10 Jun 2020

PONE-D-19-22449R2 

Co-occurrence of diabetes and depression in the U.S. 

Dear Dr. Alva:

I'm pleased to inform you that your manuscript has been deemed suitable for publication in PLOS ONE. Congratulations! Your manuscript is now with our production department. 

Kind regards, 

on behalf of

Dr. Nayu Ikeda 

Academic Editor

PLOS ONE